# The Mechanism Underlying the Hypoglycemic Effect of Epimedin C on Mice with Type 2 Diabetes Mellitus Based on Proteomic Analysis

**DOI:** 10.3390/nu16010025

**Published:** 2023-12-21

**Authors:** Xuexue Zhou, Ziqi Liu, Xiaohua Yang, Jing Feng, Murat Sabirovich Gins, Tingyu Yan, Lei Han, Huafeng Zhang

**Affiliations:** 1National Engineering Laboratory for Resource Development of Endangered Crude Drugs in Northwest China, Academician and Expert Workstations in Puer City of Yunnan Province, College of Food Engineering and Nutritional Science, Provincial Research Station of Se-Enriched Foods in Hanyin County of Shaanxi Province, International Joint Research Center of Shaanxi Province for Food and Health Sciences, Shaanxi Normal University, Xi’an 710119, Chinaguohaiwen@snnu.edu.cn (Z.L.); tyyan@snnu.edu.cn (T.Y.); lhan@snnu.edu.cn (L.H.); 2Research Station of Selenium-Enriched Tea of Shaanxi Province, Health Science Center, Xi’an Jiaotong University, Xi’an 710061, China; 3Agrarian and Technological Institute, Peoples’ Friendship University of Russia, Moscow 119991, Russia; anirr@bk.ru

**Keywords:** epimedin C, type 2 diabetes mellitus, mice, mechanism of action, label-free proteomic technique

## Abstract

Type 2 diabetes mellitus (T2DM) has become a worldwide public health problem. Epimedin C is considered one of the most important flavonoids in *Epimedium*, a famous edible herb in China and Southeast Asia that is traditionally used in herbal medicine to treat diabetes. In the present study, the therapeutic potential of epimedin C against T2DM was ascertained using a mouse model, and the mechanism underlying the hypoglycemic activity of epimedin C was explored using a label-free proteomic technique for the first time. Levels of fasting blood glucose (FBG), homeostasis model assessment of insulin resistance (HOMA-IR), and oral glucose tolerance, as well as contents of malondialdehyde (MDA) and low-density lipoprotein cholesterol (LDL-C) in the 30 mg·kg^−1^ epimedin C group (EC30 group), were significantly lower than those in the model control group (MC group) (*p* < 0.05), while the contents of hepatic glycogen, insulin, and high-density lipoprotein cholesterol (HDL-C), as well as activities of superoxide dismutase (SOD) and glutathione peroxidase (GSH-Px) in the EC30 group were notably higher than those in the MC group (*p* < 0.05). The structures of liver cells and tissues were greatly destroyed in the MC group, whereas the structures of cells and tissues were basically complete in the EC30 group, which were similar to those in the normal control group (NC group). A total of 92 differentially expressed proteins (DEPs) were enriched in the gene ontology (GO) and Kyoto Encyclopedia of Genes and Genomes (KEGG) pathways. In the EC30 vs. MC groups, the expression level of cytosolic phosphoenolpyruvate carboxykinase (Pck1) was down-regulated, while the expression levels of group XIIB secretory phospholipase A2-like protein (Pla2g12b), apolipoprotein B-100 (Apob), and cytochrome P450 4A14 (Cyp4a14) were up-regulated. According to the KEGG pathway assay, Pck1 participated in the gluconeogenesis and insulin signaling pathways, and Pla2g12b, Apob, and Cyp4a14 were the key proteins in the fat digestion and fatty acid degradation pathways. Pck1, Pla2g12b, Apob, and Cyp4a14 seemed to play important roles in the prevention and treatment of T2DM. In summary, epimedin C inhibited Pck1 expression to maintain FBG at a relatively stable level, promoted Pla2g12b, Apob, and Cyp4a14 expressions to alleviate liver lipotoxicity, and protected liver tissues and cells from oxidant stress possibly by its phenolic hydroxyl groups.

## 1. Introduction

Diabetes mellitus (DM) is a chronic disease caused by relative or absolute decreases in insulin secretion, resulting in metabolic disorders of carbohydrates, lipids, and proteins, especially an increase in blood glucose. DM is one of the worldwide public health problems. There were 537 million people who suffered from DM in 2021, and the number of patients was predicted to rise to 783 million by 2045 according to the International Diabetes Federation [1]. Traditionally, DM was divided into four types: type 1 diabetes mellitus, type 2 diabetes mellitus (T2DM), gestational diabetes mellitus, and other diabetes mellitus. Among them, T2DM persons accounted for 90~95% of the total patients [2,3]. The symptoms of T2DM are mainly hyperglycemia and lipid deposition in liver cells due to the increment in fatty acid synthesis and the reduction in fatty acid degradation [4]. Additionally, some complications such as non-alcoholic fatty liver disease (NAFLD), neuropathic diabetic retinopathy, hypertension, and cardiac disease frequently occur in T2DM patients [5,6]. In recent years, the incidence of T2DM continually increased in the world partially due to changes in dietary habits [7]. Some medicines (e.g., metformin and glimepiride) are extensively used to reduce blood glucose. However, they may cause several side effects such as gastrointestinal irritation and insulin resistance, and even exacerbate damage to patients’ pancreatic islet cells [8]. Thus, it is necessary to seek out safe and efficient chemopreventive or anti-diabetic agents.

Recently, edible medicinal plants have received widespread attention because of low side effects and drug resistance [9,10]. *Epimedium* L. (named Yin Yang Huo in Chinese), a popular edible herb in China and Southeast Asia, was listed in the Raw Materials for Health Foods by the Chinese National Health Commission (the former Ministry of Health of China) in 2002. *Epimedium* was first recorded in *Shennong’s Classic of Materia Medica* (c.a. 184–266 AD), showing efficacy in strengthening *yang*, nourishing *qi*, and improving *zhi* [11,12]. Traditionally, some Chinese herbalists have used *Epimedium* leaves to treat diabetes [13]. However, the material basis and mechanism of action remain obscure. Epimedin C, one of the important flavonoids widely distributed in various species of *Epimedium* (Appendix A) [11,12], has been demonstrated to possess pharmacological activities against glucocorticoid osteoporosis and cardiovascular diseases [14], some of which were linked to DM to some extent [15]. So, it is hypothesized that epimedin C possesses therapeutic potential against T2DM.

In order to ascertain the hypoglycemic activity of epimedin C, the effects of epimedin C on T2DM mice were systematically investigated. The influence of epimedin C on carbohydrate and lipid metabolisms, antioxidant systems, and structures of liver cells and tissues was discussed. Furthermore, the mechanism underlying the hypoglycemic effect of epimedin C on T2DM mice was explored using a label-free proteomic technique for the first time.

## 2. Materials and Methods

### 2.1. Reagents and Instruments

The main reagents included acetonitrile and formic acid (high-performance liquid chromatography (LC)/mass spectrometry (MS) grade) (Thermo Fisher Scientific Co., Ltd., Waltham, MA, USA), epimedin C (C_39_H_50_O_19_, purity ≥ 98%) (Shanghai YY Co., Ltd., Shanghai, China), glimepiride (Chongqing Conquer Pharmaceutical Co., Ltd., Chongqing, China), metformin (Bristol-Myers Squibb Co., Ltd., New York, NY, USA), streptozotocin (STZ) (Sigma-Aldrich Co., Ltd., Saint Louis, MO, USA), paraformaldehyde (Beyotime Biotech Inc., Shanghai, China), a control diet (60.0% carbohydrate, 33.0% protein, 3.0% lipid, 3.6% amino acids, 0.3% minerals and 0.1% vitamins), a high-fat diet (66.5% control diet with 20.0% sugar, 10.0% lard, 2.5% cholesterol and 1.0% bile salt) (Beijing Bo’ai Port Co., Ltd., Beijing, China), and biochemical and physiological kits (mouse insulin enzyme-linked immunosorbent assay (ELISA) kit, superoxide dismutase (SOD) kit, glutathione peroxidase (GSH-Px) kit, malondialdehyde (MDA) kit, high-density lipoprotein cholesterol (HDL-C) kit, low-density lipoprotein cholesterol (LDL-C) kit, and hepatic glycogen assay kit) (Nanjing JC Bioengineering Institute, Nanjing, China).

The main instruments included a NanoAcuity ultra-performance liquid chromatography system (Waters Technologies Co., Ltd., Milford, DE, USA) equipped with an Acclaim PepMap C_18_ peptide analysis column (Thermo Fisher Scientific Co., Ltd., Waltham, MA, USA), an Eclipse E100 light microscope equipped with a DS-U3 imaging system (Nikon Vision Co., Ltd., Tokyo, Japan), a Q Exactive Focus Hybrid Quadrupole (Q Exactive HFX) orbitrap mass spectrometer (Thermo Fisher Scientific Co., Ltd., Waltham, MA, USA), a 5804R high-speed centrifuge (Eppendorf Co., Ltd., Hamburg, Germany), and an Onetouch UltraVue glucose meter (Johnson & Johnson Co., Ltd., New Brunswick, NJ, USA).

### 2.2. Animal Experiments

#### 2.2.1. Experimental Design and Drug Administration

All animal experiments were approved by the Advisory Ethics Committee of the International Joint Research Center of Shaanxi Province for Food and Health Sciences (IJRC-A2020-001, approve date: 1 March 2020). Animal experiments were performed as previously described by Liu et al. [16] with some modifications. Four-week-old male specific pathogen-free (SPF) Kunming (KM) mice with an average weight of 18.0 ± 2.0 g were purchased from the Experimental Animal Center of Xi’an Jiaotong University (Certificate No.: SCXK (Shaan) 2012-003, Xi’an, China). The mice were raised at a temperature of 22 ± 2 °C and relative humidity of 60 ± 5% with a 12 h light/dark cycle. After adaptive feeding for seven days, all the mice were randomly divided into seven groups (*n* = 10 in each group): the NC (normal control) group, the MC (model control) group, the PC1 (the first positive control, 200 mg·kg^−1^ metformin) group, the PC2 (the second positive control, 5 mg·kg^−1^ glimepiride) group, the EC30 (30 mg·kg^−1^ epimedin C) group, the EC10 (10 mg·kg^−1^ epimedin C) group, and the EC5 (5 mg·kg^−1^ epimedin C) group (Appendix A) [14,17]. Prior to oral administration (gavage) of different drugs (metformin, glimepiride, and epimedin C) or distilled water, the mice in the PC1, PC2, EC30, EC10, EC5, and MC groups were subjected to intraperitoneal injection of STZ with citrate buffer (pH 4.5). The mice in the NC group were injected with saline instead of STZ. The mice in the PC1, PC2, EC30, EC10, EC5, and MC groups were fed a high-fat diet, while the mice in the NC group were fed the control diet. Mice with the typical symptoms of diabetes (polydipsia and polyphagia) as well as blood glucose concentrations higher than 11.1 mmol·L^−1^ for 3 consecutive days were considered DM models [18]. In the EC30, EC10, EC5, PC1, and PC2 groups, diabetic mice were supplied with different drugs, whereas mice in the NC and MC groups were provided with sterile distilled water instead of drugs for 28 days. Water and food intake, body weight, and fasting blood glucose (FBG) levels were measured at regular intervals.

#### 2.2.2. Biochemical and Physiological Analyses

##### Oral Glucose Tolerance Test

An oral glucose tolerance test was conducted according to Geidenstam et al. [19] with slight modifications. All mice were fasted overnight and supplied with glucose (2.0 g·kg^−1^ body weight) via oral administration. FBG levels were determined at 0 h (before glucose administration), 0.5 h, and 2.0 h (after glucose administration). The area under the curve (AUC) was calculated using the following formula:AUC = [(c_1_ + c_2_) × 0.5]/2 + [(c_3_ + c_2_) × 1.5]/2
where c_1_, c_2_, and c_3_ represent the FBG level (mmol·L^−1^) at 0, 0.5, and 2.0 h, respectively.

##### Assays of Insulin, Hepatic Glycogen, SOD, GSH-Px, MDA, HDL-C, and LDL-C and the Homeostasis Model Assessment of Insulin Resistance (HOMA-IR)

Samples were collected according to a previously reported protocol [16]. Briefly, the eyeballs of mice were treated to obtain blood samples, which were then centrifugated to acquire serums. Afterward, the mice were sacrificed, and their liver tissues were stored at −80 °C until needed. The levels of insulin, hepatic glycogen, MDA, HDL-C, and LDL-C as well as activities of SOD and GSH-Px, were analyzed with commercial kits according to the manufacturer’s instructions. According to Liu et al. [18], HOMA-IR was calculated using the following formula:HOMA − IR = (Sg × Si)/22.5
where Sg and Si represent the levels of blood glucose (mmol·L^−1^) and serum insulin (pmol·L^−1^), respectively.

#### 2.2.3. Histological Assessment

A histological assessment was performed using the method reported by Jack et al. [20]. In brief, fresh liver tissues were fixed with a 4% paraformaldehyde solution and then embedded in paraffin. Tissue sections (4 μm) were cut and stained with hematoxylin and eosin (H & E) stain. The structures of liver tissues and cells were observed using a light microscope (400-fold) and were photographed with an imaging system.

### 2.3. Proteomic and Bioinformatic Analyses

Proteomic analysis was carried out as previously described by Yang et al. and Liu et al. [21,22]. Briefly, proteins were extracted, quantified, alkylated, digested, and desalted, and then the peptides were obtained. The resulting peptides were ionized with a voltage of 2.0 kV and were detected with ultra-performance liquid chromatography–tandem mass spectrometry. For ultra-performance liquid chromatography, the mobile phases were composed of 0.1% aqueous formic acid (phase A) and 0.1% formic acid in acetonitrile (phase B). The elution program was as follows: 5~30% B for 110 min, 30~80% B for 5 min, 80~5% B for 0.1 min, and 5% B for 4.9 min. For mass spectrometry, an orbitrap analyzer was set within a range of 350~1600 *m*/*z* for the first scanning. Ten ions with great abundance screened with the first scanning were fragmented using higher-energy *C*-trap dissociation (HCD). In the second scanning, the orbitrap analyzer was set at 17,500 m/s. The raw files of mass spectrometry were treated with MaxQuant software (version 1.5.8.3) and the UniProt-Swissprot database. The expression levels of proteins were assessed with intensity-based absolute quantification (iBAQ) and normalized using the median normalization method [23]. Data on differentially expressed proteins (DEPs) were filtered by fold change (>1.20 or <0.83) and significance (*p* < 0.05), which were visualized as volcano plots (Appendix A). A gene ontology (GO) assay was applied to functional enrichment with the ClusterProfiler database (version 4.0), and DEPs were classified into biological processes (BP), cellular components (CC), and molecular functions (MF). The Kyoto Encyclopedia of Genes and Genomes (KEGG) database (https://www.kegg.jp/) was used to clarify the key pathways. Protein–protein interaction (PPI) networks were obtained using Cytoscape software (version 3.9.0) and STRING database (https://cn.string-db.org/) [16].

### 2.4. Statistical Analysis

All data were expressed as means ± standard deviations. Figures showing the results of the animal experiments were plotted using Origin software (version 2018), and figures showing the results of the proteomic and bioinformatic assays were plotted using the above-mentioned approaches in the section “Proteomic and bioinformatic analyses”. The statistical significance was assessed using SPSS software (version 20).

## 3. Results

### 3.1. Physiological and Biochemical Analysis

#### 3.1.1. Effects of Epimedin C on the FBG, Hepatic Glycogen, Insulin, HOMA-IR, and Oral Glucose Tolerance of T2DM Mice

As shown in Figure 1A, the FBG levels of all mice in the T2DM groups (the MC, PC1, PC2, EC30, EC10, and EC5 groups) at 0 d were significantly higher than those in the NC group (*p* < 0.05), and the FBG levels at 0 d in all T2DM groups were higher than 11.1 mmol·L^−1^, indicating that theT2DM models were successfully established before administration. From 0 to 28 d, the FBG levels of all T2DM mice were higher than those in the NC group (Figure 1B). After 14 days of treatment, the FBG levels in the MC group increased continuously, while the FBG levels of T2DM mice in the drug intervention groups (the PC1, PC2, EC30, EC10, and EC5 groups) decreased to different extents, all of which were higher than 11.1 mmol·L^−1^ (Figure 1B). At 28 d, the FBG levels of T2DM mice treated with epimedin C (the EC30, EC10, and EC5 groups) were notably lower than those in the MC group and higher than those in the NC group (*p* < 0.05). Furthermore, FBG levels exhibited a dose-dependent reduction with the increase in epimedin C concentration (Figure 1C). Compared with the FBG levels at 0 d, they decreased respectively by 27.81% in the EC30 group (*p* < 0.05), 17.14% in the EC10 group (*p* < 0.05), and 6.86% in the EC5 group (*p* > 0.05) at 28 d (Appendix A). According to Figure 1D, the hepatic glycogen content in the MC group was significantly lower than that in the NC group and the drug intervention groups (*p* < 0.05). The contents of hepatic glycogen in the drug intervention groups were significantly lower than those in the NC group (*p* < 0.05). In addition, the hepatic glycogen content in the EC30 group was 1.38 and 1.52 times higher than those in the EC10 and EC5 groups, respectively (*p* < 0.05). That is to say, the contents of hepatic glycogen increased in a dose-dependent manner with the increase in epimedin C concentration.

As seen in Figure 2A, the insulin contents in the NC group were significantly higher than those in the MC group (*p* < 0.05), while there was no significant difference in the insulin contents between the NC, PC1, PC2, and EC30 groups (*p* > 0.05) (Appendix A). Moreover, the HOMA-IR levels in the MC group were noticeably higher than those in the NC and EC groups (*p* < 0.05) (Figure 2B and Appendix A).

Figure 3A illustrates the effects of epimedin C on the oral glucose tolerance of T2DM mice. At 0 h (before glucose gavage), the FBG levels in the MC group were significantly higher than those in the drug intervention groups and the NC group (*p* < 0.05), and the FBG levels of T2DM mice were remarkably higher than that in the NC group (*p* < 0.05). However, there were no significant differences in the FBG levels between the PC1, PC2, and EC30 groups (*p* > 0.05). The FBG levels in mice in all groups rose at 0.5 h (after glucose gavage), whereas the FBG levels in the NC and EC30 groups decreased at 2.0 h (after glucose gavage). According to Figure 3B, the AUC in the MC group was significantly higher than that in the NC group and the drug intervention groups (*p* < 0.05). In particular, the AUC decreased in a dose-dependent manner with the increase in epimedin C concentration.

#### 3.1.2. Effects of Epimedin C on Oxidative Stress

The MDA contents in the MC group were significantly higher than those in the NC group (*p* < 0.05) in the serums and livers, whereas the activities of SOD and GSH-Px were markedly lower than those in the NC group (*p* < 0.05) (Figure 4). The activities of SOD and GSH-Px in the PC1, PC2, and EC30 groups were significantly higher than those in the MC group (*p* < 0.05) (Figure 4A,C,D,F), while the MDA contents in the PC1, PC2, and EC30 groups were notably lower than those in the MC group (*p* < 0.05) (Figure 4B,E). In particular, the GSH-Px activity in livers and serums rose in a dose-dependent manner with the improvement in epimedin C concentration.

#### 3.1.3. Effects of Epimedin C on Blood Lipid Levels

As shown in Figure 5, the HDL-C contents in the serum and livers of mice in the MC group were significantly lower than those in the NC group (*p* < 0.05), while the LDL-C contents were significantly higher than those in the NC group (*p* < 0.05). The HDL-C contents in the PC1 and EC30 groups were notably higher than that in the MC group (*p* < 0.05), while the LDL-C contents in the PC1 and EC30 groups were markedly lower than that in the MC group (*p* < 0.05) (Figure 5). There were no significant differences in the HDL-C and LDL-C contents between the EC30 and NC groups in the serums (*p* > 0.05). Similarly, the difference in the LDL-C content between the EC30 and NC groups in the livers was not significant (*p* > 0.05). These results indicated that epimedin C dose-dependently elevated HDL-C contents and reduced LDL-C contents in livers and serums.

### 3.2. Effects of Epimedin C on Liver Histopathological Changes

Figure 6 illustrates the impacts of epimedin C on liver tissues and cells. In the NC group, the liver tissues had a complete structure with an orderly arrangement of hepatic lobules, hepatic cords, and hepatic sinusoids. In addition, the cytoplasm of hepatocytes was homogeneous, and no cytoplasmic granules, oil droplets, or fibrous tissues were observed. Conversely, the structures of liver cells and tissues were greatly destroyed in the MC group, and vacuolar degeneration was found in hepatocytes. Some hepatocytes became oedematous, and some hepatocytes were surrounded with lipid droplets. In particular, the outlines of some hepatocytes became blurred, and the arrangement of hepatic cords and sinusoids became irregular (Figure 6B). Compared with the MC group, the structures of liver cells and tissues in each drug intervention group were basically complete with less vacuolar degeneration of hepatocytes and a more regular arrangement of hepatocyte cords and sinusoids (Figure 6C–G). In the EC30 group, the structures of liver cells and tissues were similar to those in the NC group (Figure 6G), which demonstrated the therapeutic effect of epimedin C on liver damage induced with T2DM.

### 3.3. Proteomic Analysis

#### 3.3.1. Identification of DEPs

According to the results of the animal experiments, the levels of FBG, HOMA-IR, and oral glucose tolerance, as well as the MDA and LDL-C contents, in the EC30 group were significantly lower than those in the MC group, while the hepatic glycogen, insulin, and HDL-C contents, as well as the activities of SOD and GSH-Px in the EC30 group, were notably higher than those in MC group. In order to further ascertain the hypoglycemic effect of epimedin C on diabetic mice and to reveal the mechanism of action of epimedin C against T2DM, liver tissues of mice in the NC, MC, and EC30 groups were subjected to proteomic analysis. Uniprot-Swissprot was used for a database search and value analysis, and 46,168 matched spectra were obtained, which were assigned to various peptides, proteins, and/or enzymes. In the NC group vs. the MC group, a total of 324 DEPs were found, which included 161 down-regulated proteins and 163 up-regulated proteins. In the EC30 group vs. the MC group, 331 DEPs, including 148 down-regulated proteins and 183 up-regulated proteins were identified (Figure 7A and Appendix A). A Venn diagram was plotted, and 139 common DEPs were identified between the NC group and the MC group and the EC30 group and the MC group (Figure 7B). These 139 DEPs were used for the hierarchical clustering analysis (Figure 7C). According to the heatmap, the EC30 and NC groups belonged to the same cluster, while the MC group was separated from them (Figure 7C), illustrating that the expression profile of DEPs in the EC30 group was similar to that in the NC group. The remarkable similarity in the expression profile of DEPs in the EC30 and NC groups implied that epimedin C had a therapeutic effect on T2DM mice. In the boxed parts of the heatmap, there were 92 DEPs (54 up-regulated proteins and 38 down-regulated proteins) in the EC30 group vs. the MC group (Figure 7C), which might be the key proteins during T2DM treatment with epimedin C.

#### 3.3.2. GO and KEGG Pathway Analyses

The GO and KEGG pathway assays involving 92 key DEPs are presented as histograms (Figure 8). According to the GO analysis, several proteins were intimately linked to GO functions such as CC, MF, and BP (Appendix A). These functional proteins included acetyl-CoA carboxylase 2 (Acacb), diazepam-binding inhibitor (Dbi), cytochrome P450 4A14 (Cyp4a14), threonine synthase-like 2 (Thnsl2), peroxisomal acyl-coenzyme A oxidase 2 (Acox2), glutamate oxaloacetate transaminase 1 (Got1), methionine adenosyltransferase 1 alpha (Mat1a), cytosolic phosphoenolpyruvate carboxykinase (Pck1), etc. (Appendix A). Regarding CC, DEPs were mainly related to protein–lipid complexes, secondary lysosomes, high-density lipoprotein particles, and the perinuclear endoplasmic reticulum, most of which contributed to lipid anabolism (Figure 8A). Regarding MF, DEPs were mainly correlated with carboxylic acid binding, organic acid binding, fatty acid derivative binding, oxidoreductase activity, and carboxy-lyase activity. Regarding BP, DEPs took part in fatty acid metabolism, carboxylic acid and organic acid anabolism or catabolism, and lipid catabolism (Figure 8A). According to the KEGG pathway analysis, 24 pathways were enriched (Figure 8B). Among them, seven pathways were directly involved in the metabolisms of fatty acids and their derivatives, namely, the peroxisome proliferator-activated receptor (PPAR) signaling pathway, arachidonic acid metabolism, fatty acid metabolism, α-linolenic acid metabolism, unsaturated fatty acid biosynthesis, fatty acid degradation, and fat digestion and absorption (Figure 8B). In contrast, one pathway (i.e., pyruvate metabolic pathway) was indirectly related to fatty acid anabolism through the synthesis of intermediates (acetyl CoA). There were 10 DEPs annotated in KEGG pathways directly related to fatty acid metabolism, including Dbi, Cyp4a14, Acox2, Pck1, fatty acid desaturase 1 (Fads1), fatty acid desaturase 2 (Fads2), group XIIB secretory phospholipase A2-like protein (Pla2g12b), apolipoprotein B-100 (Apob), prostaglandin E synthase 3 (Ptges3), and aldehyde dehydrogenase family 3 member A2 (Aldh3a2). To sum up, a total of 14 proteins were identified with the GO and KEGG pathway analyses.

#### 3.3.3. DEPs Network Interaction Analysis

To understand the PPI networks of the key proteins, 92 DEPs were screened with the hierarchical clustering analysis and 14 DEPs characterized with GO and KEGG assays were analyzed using the SRTING tool. As shown in Figure 9A,B, the interaction score of DEPs varied greatly, and some proteins exhibited a high degree of protein–protein connectivity. For instance, Pla2g12b was closely associated with Fads2, Apob, Cyp4a14, and Pck1. In the EC30 group vs. the MC group, some proteins were up-regulated, such as Pla2g12b, Cyp4a14, and Apob. The expression levels of some proteins were down-regulated, such as Pck1 (Figure 9B). Pck1, Apob, Pla2g12b, and Cyp4a14 had a high degree of protein–protein connectivity in the PPI networks, which might be the key proteins involved in the PPAR signaling pathway, gluconeogenesis, fatty acid degradation, fat digestion and absorption, and arachidonic acid metabolism. According to the DEP network interaction analysis, epimedin C alleviated T2DM and regulated lipid metabolism partially by increasing the expression levels of Pla2g12b, Cyp4a14, and Apob and decreasing the expression level of Pck1 in the livers of T2DM mice.

## 4. Discussion

Normally, insulin secreted by pancreatic β-cells maintains FBG at a relatively stable level mainly by promoting hepatic glycogen synthesis and inhibiting gluconeogenesis [24]. However, in diabetic mice, glucose metabolism in vivo is most probably disturbed partially due to pancreatic β-cells destruction and insulin secretion shortage. Thus, FBG might reach a high level, hepatic gluconeogenesis might be hampered, and symptoms like insulin resistance and oral glucose tolerance might appear [25]. Once insulin resistance takes place, not only does insulin no longer inhibit gluconeogenesis, but it also stimulates hepatic fat synthesis, both of which exacerbate insulin resistance and other symptoms [26]. After T2DM mice were treated with epimedin C in the present work, FBG, HOMA-IR, and oral glucose tolerance significantly decreased, while the hepatic glycogen and insulin contents remarkably increased. It is suggested that epimedin C improves the secretion function of pancreatic β-cells, attenuates insulin resistance, increases serum insulin contents, promotes hepatic glycogen synthesis, and keeps FBG nearly constant in T2DM mice.

It was reported that oxidative stress in vivo is closely related to the occurrences of T2DM and its complications [27]. T2DM patients with chronic hyperglycemia are prone to accumulate advanced glycosylation end products (AGEs), which enhance the activity of nicotinamide adenine dinucleotide phosphate (NADPH) oxidase and increase the amount of reactive oxygen species (ROS) [28,29]. Excessive ROS might attack the cytoplasmic membrane, trigger cellular damage and dysfunction of organelles such as mitochondria and endoplasmic reticulum, and exacerbate lipid peroxidation [30], which was regarded as the major cause of NAFLD [31]. In this study, the SOD and GSH-Px activity in the livers and serums of T2DM mice treated with epimedin C in the EC30 group were significantly higher than those in the MC group (*p* < 0.05) but lower than those in the NC group. Moreover, the MDA content in the EC30 group was remarkably lower than that in the MC group (*p* < 0.05) but did not differ significantly from that in the NC group (*p* > 0.05). It is known that SOD and GSH-Px have evident antioxidant activity, which can maintain the homeostasis of in vivo antioxidant system by scavenging ROS. And MDA is the product of intracellular lipid peroxidation. Numerous reports indicated that phenolic hydroxyl groups in the chemical structures of flavonoids could react with free radicals to reduce ROS in the organism and to mitigate damage to tissues and cells resulting from peroxidation [32]. The effects of epimedin C on SOD and GSH-Px activities and MDA contents demonstrated its protective ability against oxidant stress related to T2DM, which was possibly attributed to its phenolic hydroxyl groups.

Because the occurrence and development of T2DM are usually accompanied by metabolic disorders of lipid, intervention of lipid metabolism plays an important role in the prevention and treatment of T2DM. Cholesterol is mainly transported in the blood using LDL-C, which enters cells after specific binding with low-density lipoprotein receptor (LDLR) on peripheral cell membranes and implements its physiological function after hydrolysis by lysosomes [33]. Excessive LDL-C will be phagocytosed by macrophages on vascular walls and then be hydrolyzed to free cholesterol, which may bind with high-density lipoprotein (HDL) to form HDL-C. The resulting HDL-C may be ingested by liver cells via scavenger receptor class B type 1 (SR-B1) [34]. Normally, cholesterol ingested by the liver can be excreted through bile or converted to very low-density lipoprotein cholesterol (VLDL-C), which subsequently re-enters blood circulation [35]. However, T2DM patients’ capacities to transport HDL-C are frequently low. When cholesterol level exceeds the patients’ capacities, HDL-C levels may decrease while LDL-C levels increase [36], which form an important pathological basis for vascular atherosclerosis complicated by diabetes [37]. In this study, the HDL-C contents in the livers and serums of mice in the EC30 group were significantly higher than those in the MC group, and the LDL-C contents in the EC30 group were notably lower than those in the MC group. These results demonstrated the regulatory effects of epimedin C on lipid metabolism disorders.

The pathogenesis of T2DM is very complex and involves numerous proteins and pathways [38]. Consequently, the label-free proteomic technique was used to clarify the key pathways and identify the important proteins that were differentially expressed among various groups in the presence or absence of epimedin C in the present study. Notably, epimedin C was found to alleviate the abnormalities of lipid metabolism in T2DM mice mainly by regulating the expression levels of Pck1, Pla2g12b, Apob, and Cyp4a14. Pck1 is a rate-limiting enzyme in gluconeogenesis, which is sometimes regarded as a candidate target of anti-diabetic agents [39]. It was reported that liver-specific overexpression of Pck1 in mice resulted in FBG elevation, lipid deposition, and insulin resistance [40]. In this work, the expression level of Pck1 in the EC30 group was close to that in the NC group. Nevertheless, its expression level in the EC30 group was significantly lower than that in the MC group. Based on the KEGG pathway analysis in the EC30 group vs. the MC group, Pck1 was observed to take part in pyruvic acid metabolism, glycolysis, gluconeogenesis, and insulin signaling pathways (Appendix A). These results implied that epimedin C treated T2DM partly by inhibiting Pck1 expression, regulating glycolysis, promoting hepatic glycogen synthesis, and activating insulin signaling pathways. Apob and Pla2g12b play important roles in the assembly and secretion of hepatic low-density lipoproteins (VLDL), which are transcriptionally regulated by hepatocyte nuclear factor-4α (HNF-4α) [41]. Mice with knockouts of Apob and Pla2g12b genes secreted less VLDL but accumulated more triglycerides, cholesterol, and fatty acids in their livers, which frequently led to NAFLD [42,43]. In this study, the expression levels of Pla2g12b and Apob in the MC group were significantly lower than those in the NC group, while the expression levels of these two proteins in the EC30 group were close to those in the NC group. The GO and KEGG pathway assays indicated that Pla2g12b and Apob were co-enriched in lipolytic metabolism as well as fat digestion and absorption pathways in the EC30 group vs. the MC groups (Figure 8 and Appendix A). These results suggested that epimedin C could increase the expression levels of Pla2g12b and Apob, promote the secretion of liver VLDL, facilitate lipolysis as well as fat digestion and absorption, reduce the deposition of lipid in livers, and thus alleviate the symptoms of T2DM. Cytochrome P450 (Cyp) epoxygenases can catalyze the conversion of arachidonic acid into other bioactive eicosanoids. Cyp4a14 is one of the important Cyp epoxygenases, which catalyze ω-hydroxylation of long-chain and medium-chain fatty acids such as arachidonic acid [44]. It was reported that the expression level of Cyp4a14 in obese mice was significantly lower than that in normal mice [45]. After treating obese mice with the lipid-lowering drug WY-14643, Cyp4a14 expression notably up-regulated, and lipid deposition remarkably reduced [46]. In this research, the expression level of Cyp4a14 in the MC group was significantly lower than that in the NC group, while its expression level in the EC30 group was close to that in the NC group. The GO and KEGG pathway assays also revealed that Cyp4a14 was enriched in fatty acid degradation pathways, especially arachidonic acid catabolism (Figure 8 and Appendix A). These results suggested that epimedin C could increase the expression level of Cyp4a14, inhibit excessive lipid deposition, and protect the liver from lipotoxicity, all of which might contribute to the prevention and treatment of T2DM.

## 5. Conclusions

In this study, the therapeutic potential of epimedin C against T2DM was confirmed using an animal model, and the mechanism underlying the hypoglycemic activity of epimedin C was determined using the label-free proteomic technique for the first time. Notably, intervention with 30 mg·kg^−1^ epimedin C decreased levels of FBG, HOMA-IR, and oral glucose tolerance, as well as the contents of MDA and LDL-C in T2DM mice, while it increased the hepatic glycogen, insulin, and HDL-C contents, as well as the activities of SOD and GSH-Px. In addition, epimedin C effectively protected the structures of liver cells and tissues. There were 92 key DEPs enriched in the GO and KEGG pathways. In the EC30 group vs. the MC group, the expression level of Pck1 was down-regulated, while expression levels of Pla2g12b, Apob, and Cyp4a14 were up-regulated. According to the KEGG pathway assay, Pck1 participated in gluconeogenesis and the insulin signaling pathways, and Pla2g12b, Apob, and Cyp4a14 were the key proteins in fat digestion and the fatty acid degradation pathways. Pck1, Pla2g12b, Apob, and Cyp4a14 seemed to play important roles in the prevention and treatment of T2DM. In summary, epimedin C could inhibit Pck1 expression to maintain FBG at a relatively stable level, promote Pla2g12b, Apob, and Cyp4a14 expressions to alleviate liver lipotoxicity, and protect liver tissues and cells from oxidant stress possibly by its phenolic hydroxyl groups.

## Figures and Tables

**Figure 1 nutrients-16-00025-f001:**
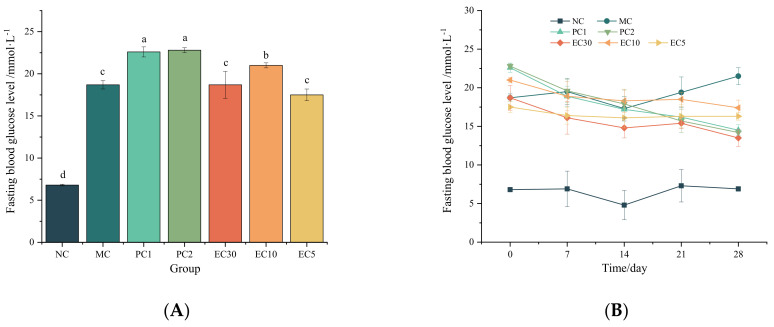
Effects of epimedin C on the fasting blood glucose (FBG) level and the hepatic glycogen content. (**A**) FBG levels at 0 d (before administration); (**B**) changes in FBG levels from 0 to 28 d; (**C**) FBG levels at 28 d; (**D**) hepatic glycogen content at 28 d. Different letters on the top of columns indicate significant differences between groups (*p* < 0.05).

**Figure 2 nutrients-16-00025-f002:**
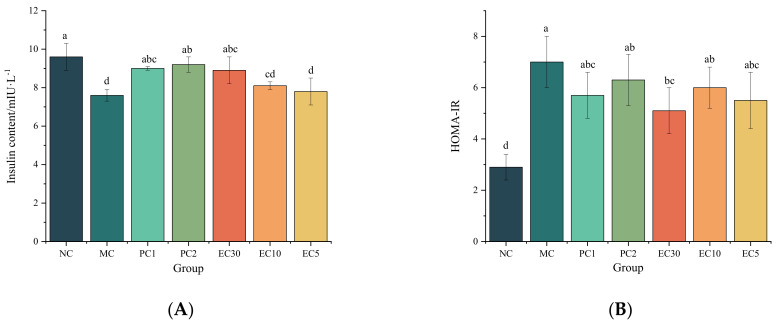
Effects of epimedin C on insulin content in serum (**A**) and homeostasis model assessment of insulin resistance (HOMA-IR) (**B**). Different letters on the top of columns indicate significant differences between groups (*p* < 0.05).

**Figure 3 nutrients-16-00025-f003:**
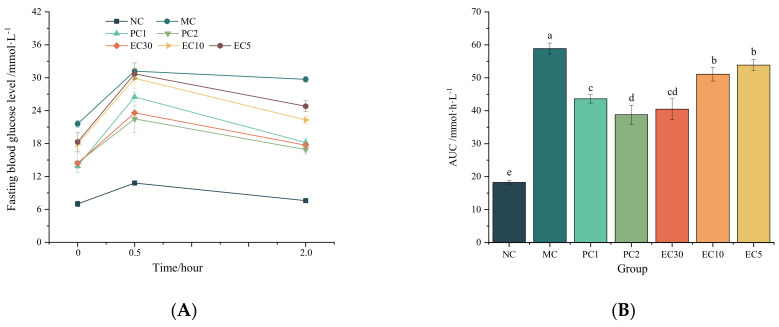
Effects of epimedin C on oral glucose tolerance. (**A**) FBG levels from 0 to 2 h and (**B**) area under the curve (AUC). Different letters on the top of columns indicate significant differences between groups (*p* < 0.05).

**Figure 4 nutrients-16-00025-f004:**
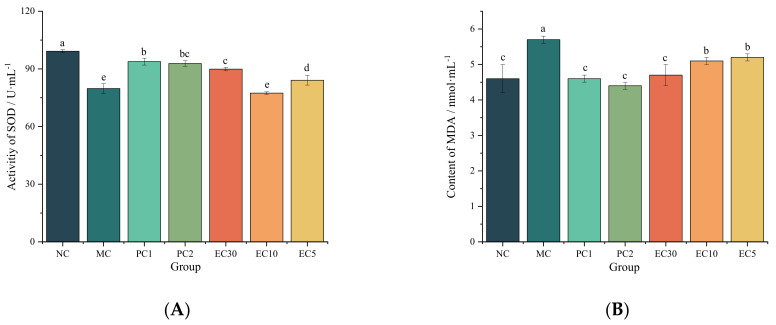
Effects of epimedin C on oxidative stress. (**A**) Superoxide dismutase (SOD) activity in serum; (**B**) malondialdehyde (MDA) content in serum; (**C**) glutathione peroxidase (GSH-Px) activity in serum; (**D**), SOD activity in livers; (**E**) MDA content in livers; (**F**) GSH-Px activity in livers. Different letters on the top of columns indicate significant differences between groups (*p* < 0.05).

**Figure 5 nutrients-16-00025-f005:**
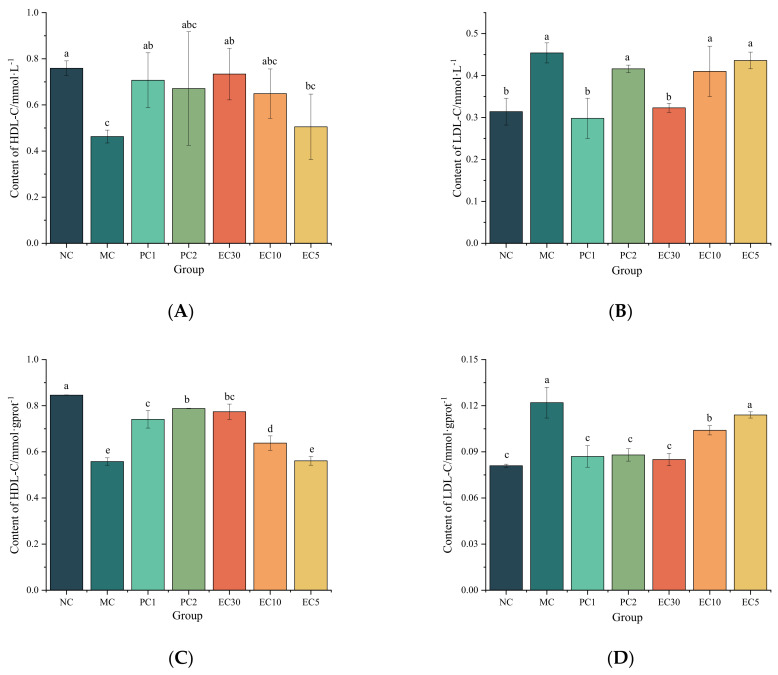
Effects of epimedin C on blood lipid levels. (**A**) High-density lipoprotein cholesterol (HDL-C) content in serums; (**B**) low-density lipoprotein cholesterol (LDL-C) content in serums; (**C**) HDL-C content in livers; (**D**), LDL-C content in livers. Different letters on the top of columns indicate significant differences between groups (*p* < 0.05).

**Figure 6 nutrients-16-00025-f006:**
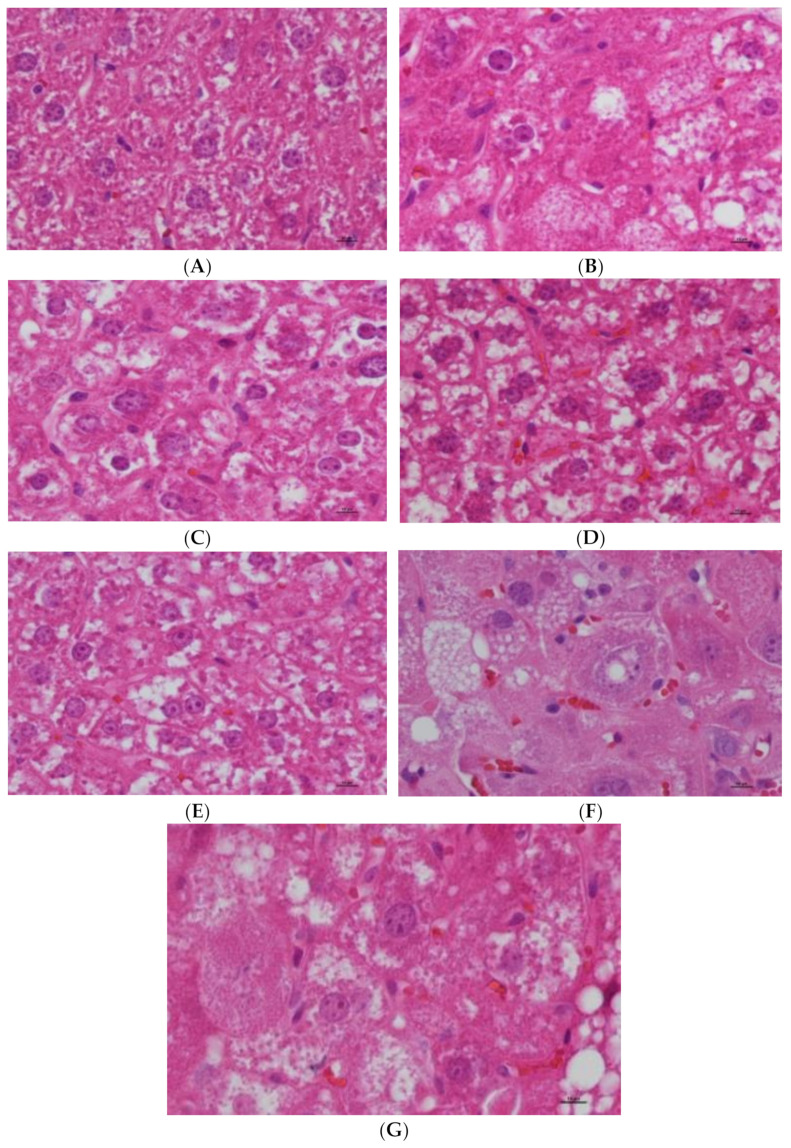
Effects of epimedin C on liver histopathological changes. Structures of liver cells and tissues in the NC group (**A**), the MC group (**B**), the PC1 group (**C**), the PC2 group (**D**), the EC30 group (**E**), the EC10 group (**F**), and the EC5 group (**G**).

**Figure 7 nutrients-16-00025-f007:**
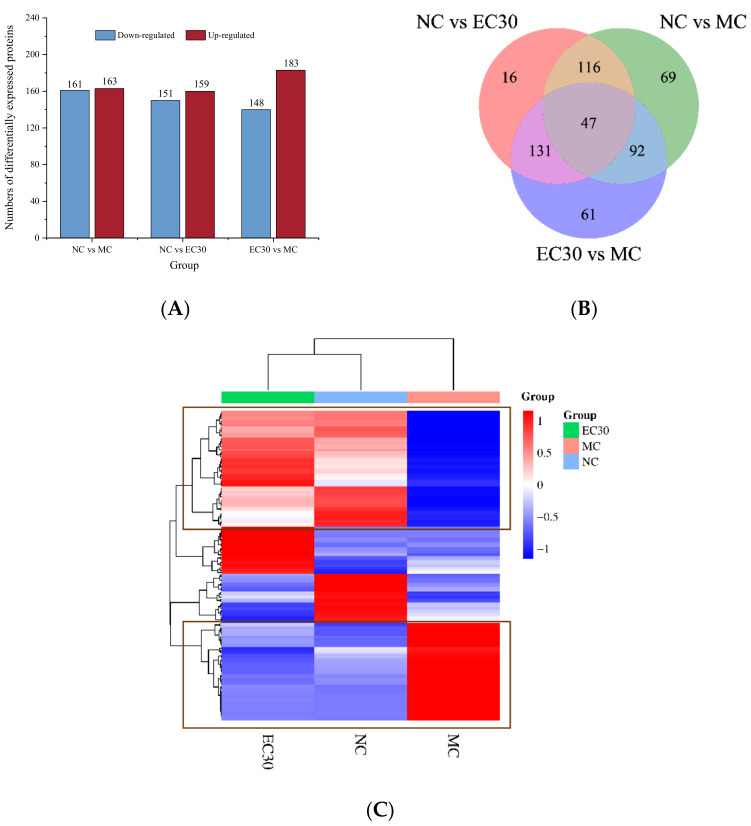
Differentially expressed protein (DEP) analyses in different groups. (**A**) The numbers of down-regulated and up-regulated DEPs in the NC group vs. the MC group, the NC group vs. the EC30 group, and the EC30 group vs. the MC group. (**B**) Venn diagram of DEPs in the NC group vs. the MC group, the NC group vs. the EC30 group and the EC30 group vs. the MC group. (**C**) Hierarchical clustering analysis of DEPs in the NC, MC, and EC30 groups.

**Figure 8 nutrients-16-00025-f008:**
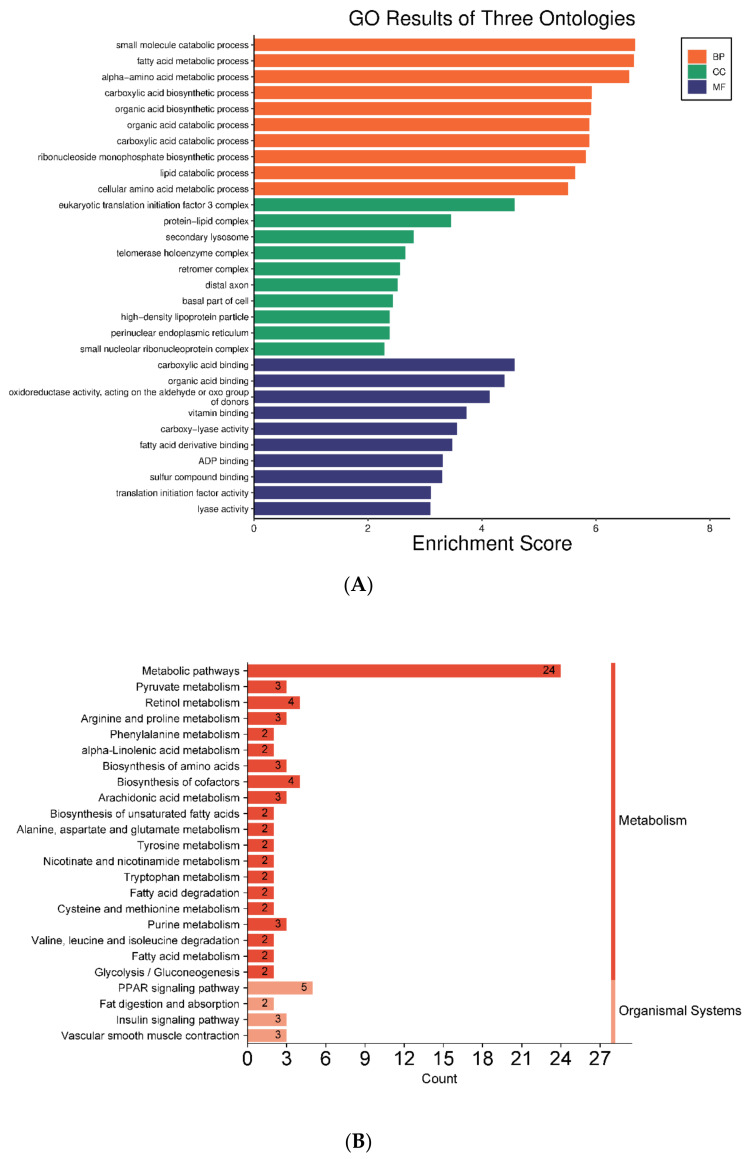
Gene ontology (GO) (**A**) and Kyoto Encyclopedia of Genes and Genomes (KEGG) pathway analyses (**B**) of DEPs.

**Figure 9 nutrients-16-00025-f009:**
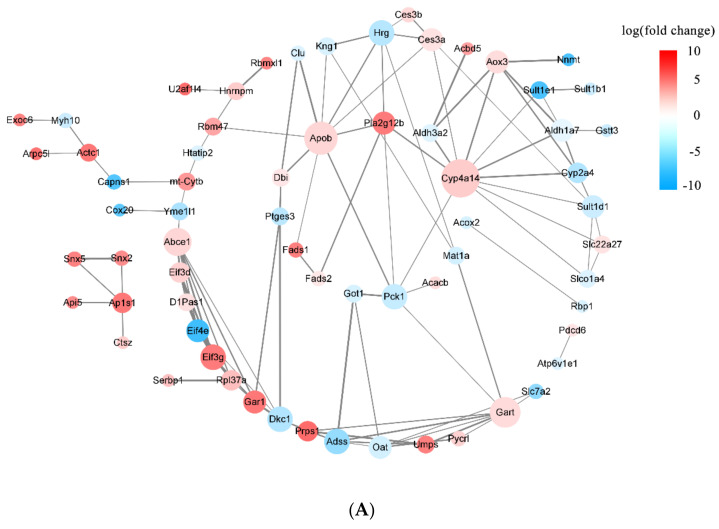
Protein–protein interaction (PPI) networks and degree of protein–protein connectivity of DEPs. (**A**) PPI network of 92 DEPs screened with a hierarchical clustering analysis. (**B**) PPI network of 14 DEPs characterized with GO and KEGG assays. Red nodes represent up-regulated DEPs, and blue nodes represent down-regulated DEPs. The size of each node reflects the degree of interaction. Line thickness indicates the interaction score between two DEPs, and only DEPs with an interaction score >0.4 are displayed.

## Data Availability

All data generated or analyzed during this study are included in this publication and the Appendix A.

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
