# Peer review of "The Mechanism Underlying the Hypoglycemic Effect of Epimedin C on Mice with Type 2 Diabetes Mellitus Based on Proteomic Analysis"

_nutrients, 2023, doi:10.3390/nu16010025_

Round 1
Reviewer 1 Report
Comments and Suggestions for Authors
The article “Mechanism underlying hypoglycemic effect of epimedin C on
mice with type 2 diabetes mellitus via proteomic analysis” by Zhou and colleagues reports on the results of an experiment conducted on diabetic mice to compare the effects of the flavonoid epidemin C, and two standard pharmacological treatments for diabetes, on several biomarkers associated with the disease.
I have the following questions and comments for the authors:
Line 13, “It is of theoretical and practical significance to screen natural nutraceuticals or phytopharmaceuticals for preventing and treating T2DM from edible medicinal plants”: I think this sentence is misleading and should be removed from the abstract. Indeed, here you are not studying the effects of a medicinal plant eaten whole or fresh or at least in a dietary approach, but the effects of a single molecule extracted from it, used no differently than any other drug. I would replace with: “Type 2 diabetes mellitus (T2DM) has become a worldwide public health problem. Epimedin C is one of the most important flavonoids in Epimedium, a famous edible herb in China and Southeast Asia, traditionally used in herbal medicine to treat diabetes. In the present study…”
Line 62, Epimedium: please provide the taxonomic name of the plant
Line 69, Epimedin C, please provide structure of this molecule in a figure
Line 83, epimedin C: was it extracted from epimedium, or synthetic?
How much epidemin C is present in Epidemium? To how much of the fresh herb would the dosage given to mice correspond, to have the same amount of the molecule?
Line 104. Since the experimental design is complex, a graphical depiction of the seven groups with the indication of the different STZ treatment, different diet and different drug received would be helpful
Line 120. Why did the the non-diabetic mice get a completely lipid-free diet? Couldn’t they eat the same diet as the diabetic animals? What is the purpose of having a control group if they follow a different diet?
Line 127, please also report the duration of the treatment here, and the number of mice in each group
Figure 1D, please specify in the caption after how many days of treatment the data refer to
Line 199, “Compared with FBG levels at 0 d, they decreased respectively by 27.81% in EC30 group, 17.14% in EC10 group, and 6.86% in EC5 group at 28 d”. Please indicate whether these variations are statistically significant. Also, it is not very easy to observe these decreases in figure 1C. Wouldn’t it be better to combine Figure 1A and 1C, so that the pre and post bars for each group are next to each other?
Line 268, Just showing selected pictures from the different groups is not really representative, especially since if you only present qualitative obervations in the text. There are several scoring systems to translate histology images into numerical data that you can use for a more accurate statistical analysis.
Comments on the Quality of English Language
The English language needs major revision. There are a lot of mistakes and imprecisions (such as “There were 537 million people suffered from DM in 2021”, line 46; “T2DM persons”, line 50, “which involved in numerous proteins”, line 440…) that currently make it difficult to read.
Author Response
Dear editor,
First of all, we are grateful to you for your kind work. We have revised our manuscript according to you and the reviewers. The comments and our answer are listed as an attachment.
In the revised version of our manuscript, all corrections or revisions have been highlighted with red font. If you have any queries about our manuscript, please do not hesitate to contact me. Thank you again for your kind work.
Best regards!
Sincerely yours,Hua-Feng Zhang (Director, PI, Chair Professor, Ph.D.)
Director and principal investigator (PI) of International Joint Research Center of Shaanxi Province for Food and Health Sciences
Chair professor and chief expert of Provincial Expert Workstation for Hua-Feng Zhang, Academician and Expert Workstation in Puer City of Yunnan Province
Shaanxi Normal University, Xi’an City, Shaanxi Province, P.R. China
Website: http://ijrc.snnu.edu.cn/info/1014/1066.htm
E-mail: isaacsau@sohu.com; ijrc@snnu.edu.cn

Reviewer 2 Report
Comments and Suggestions for Authors
Review for the article “Mechanism underlying hypoglycemic effect of epimedin C on mice with type 2 diabetes mellitus via proteomic analysis”
In this study the authors wanted to study the potential effect of epimedin C against T2DM using animal model. They found that 30 mg·kg-1 epimedin C decreased levels of FBG, HOMA-IR and oral glucose tolerance, as well as contents of MDA and LDL-C in T2DM mice, while increased contents of hepatic glycogen, insulin and HDL-C, as well as activities of SOD and GSH-Px. Also, epimedin C promote Pla2g12b, Apob and Cyp4a14 expressions and protect liver tissues and cells from oxidant stress.
Comments
How did they determine the dose of metformin, glimepiride, epimedin C, that were administered? Did they performed a subclinical toxicity study at the beginning to determine the dose to be administered?
Did the authors perform a dose-effect study?
The authors said that the drugs were administered orally. Please give details.
Line 188: The authors said “FBG levels of all mice in T2DM groups (MC, PC1, PC2, EC30, 188 EC10 and EC5 groups) at 0 d were significantly higher than those in NC group”. Also, in line 213 the authors said “insulin contents in NC group were significantly higher than 213 those in MC group (P<0.05), while there was no significant difference in insulin contents 214 between NC, PC1, PC2 and EC30 groups (P>0.05).”There are more examples in the article. Please give the value for these parameters in a table.
Author Response
Dear editor,
First of all, we are grateful to you for your kind work. We have revised our manuscript according to you and the reviewers. The comments and our answer are listed as an attachment.
In the revised version of our manuscript, all corrections or revisions have been highlighted with red font. If you have any queries about our manuscript, please do not hesitate to contact me. Thank you again for your kind work.
Best regards!
Sincerely yours,Hua-Feng Zhang (Director, PI, Chair Professor, Ph.D.)
Director and principal investigator (PI) of International Joint Research Center of Shaanxi Province for Food and Health Sciences
